# The Asymmetric Petasis Borono-Mannich Reaction: Insights on the Last 15 Years

Carolina Marques [1,*] and Pedro Brandão [2,3,4,5,*]

1   LAQV-REQUIMTE, Institute for Research and Advanced Studies, University of Évora,
    Rua Romão Ramalho, 59, 7000-641 Évora, Portugal
2   Egas Moniz Interdisciplinary Research Center (CiiEM), Egas Moniz School of Health and Science,
    Quinta da Granja, Monte da Caparica, 2829-511 Caparica, Portugal
3   iBB-Institute for Bioengineering and Biosciences, Department of Bioengineering, Instituto Superior Técnico,
    University of Lisboa, 1049-001 Lisbon, Portugal
4   Associate Laboratory i4HB–Institute for Health and Bioeconomy at Instituto Superior Técnico,
    University of Lisboa, Av. Rovisco Pais, 1049-001 Lisbon, Portugal
5   CQC-IMS, Department of Chemistry, University of Coimbra, Rua Larga, 3004-535 Coimbra, Portugal
*   Correspondence: carolsmarq@uevora.pt (C.M.); pbrandao@egasmoniz.edu.pt (P.B.)

**Abstract:** The Petasis borono-Mannich reaction, commonly described as the Petasis reaction, was one of the latest famous multicomponent reactions described in the literature. Currently celebrating its 30th anniversary since it was first reported by Petasis and Akritopoulou in 1993, this reaction has emerged as a powerful tool for the synthesis of biologically relevant molecules (such as substituted amines or amino acids), among others. This three-component catalyst-free reaction (the classic model), involving the coupling of an aldehyde, an amine, and a boronic acid, enables the synthesis of polysubstituted amine-containing molecules. Several accounts regarding the catalyst-free version using different carbonyl, amine, and boron-type components have been reported thus far. In contrast, the asymmetric version is still in its infancy since it was first reported in 2007. In this work, we aim to review the asymmetric versions of the Petasis reaction reported over the last 15 years, considering the chiral pool approach (asymmetric induction by one reaction component) and the use of catalysts (organocatalysts, transition-metal catalysts, and others) to access enantiomeric and diastereomeric pure amino-derivatives. Insights regarding the catalyzed Petasis reaction and consequent sustainable synthesis will be highlighted.

**Keywords:** Petasis borono-Mannich reaction; asymmetric; enantioselective; diastereoselective; organocatalyst; multicomponent reaction

## 1. Introduction

Multicomponent reactions (MCRs) can be defined as reactions in which three or more components are combined, simultaneously, in one reaction vessel, leading to a final product that contains most of the atoms present in the starting reagents. Some of the inherent advantages of the use of MCRs are easy access to structural diversity and the ability to attain small molecule libraries in a time-efficient manner. These features make MCRs a suitable approach for drug discovery and development. Proof of that can be found in the literature regarding the synthesis of several bioactive molecules and active pharmaceutical ingredients (APIs) using MCRs [1–4]. Furthermore, the reduction of reaction steps, work-up procedures, and overall high yields make MCRs highly compliant with green chemistry principles [5]. The Petasis MCR, also known as Petasis borono-Mannich MCR, has showcased its applicability for drug discovery and development since it was first reported by Nicos A. Petasis and I. Akritopoulou in 1993 [6]. In their work, the synthesis of the topical antifungal agent naftifine was successfully achieved using this approach. Since then, several other APIs have been synthesized using the Petasis MCR, including fingolimod [7], zanamivir [8], and acalabrutinib [9] (Figure 1).

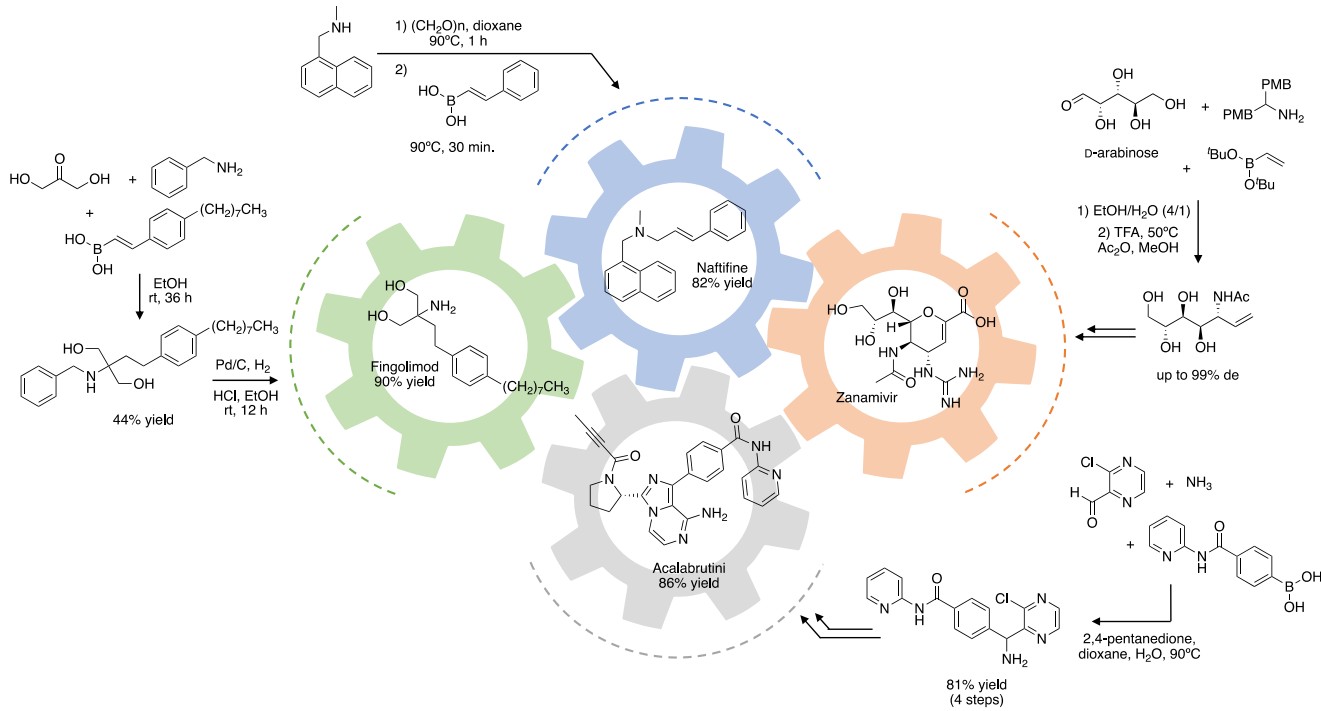

**Figure 1.** Examples of APIs synthesized using the Petasis MCR.

Mechanistically, the Petasis MCR undergoes several reversible steps in equilibrium, ending in an irreversible step which leads to the reaction to completion. Although the full reaction mechanism remains undisclosed, during the past three decades since this reaction was first reported, several efforts based on experimental and theoretical data indicate that an iminium ion is formed by reacting the aldehyde and the amine component (to illustrate, a salicylaldehyde derivative and a secondary amine were selected as examples in Scheme 1). Then, the phenol functional group activates the boronic acid, generating a tetrahedral boronate salt intermediate (the "ate complex"), which is capable of transferring the boron substituent to the iminium moiety, affording the Petasis adduct [10–12].

**Scheme 1.** Petasis MCR mechanism.

One of the main advantages of the Petasis MCR, even when compared to several other MCRs, is its suitability to undergo catalytic asymmetric transformations, which is of great interest for drug discovery and API synthesis [13,14]. Enantioselectivity can be induced through the use of reagents from the chiral pool or asymmetric catalysts. In this review, we aim to exhibit the recent efforts reported for the asymmetric Petasis MCR and how useful this reaction is to unlock new chiral compounds.

## 2. Asymmetric Induction by Reaction Components (Chiral Pool)

Using reagents from the chiral pool is a prominent approach to synthesizing enantiomerically pure bioactive molecules with diverse levels of structural complexity. These

reagents are often affordable, are commercially available, and enable synthetic protocols which require milder reaction conditions than those needed to operate with several asymmetric catalysts [15,16].

## 2.1. Carbonyl Component

The influence of the carbonyl component in the Petasis MCR is one of the most widely explored. Usually, the presence of a hydroxyl or carboxylic acid group in the proximity of the reacting carbonyl group is a requirement for the activation of the organoboronic acid as an "ate complex" (see Scheme 1). Among the studied possibilities, α-substituted chiral aldehydes constitute a solid approach for obtaining enantioenriched Petasis adducts. Thaima and Pyne explored the use of (*S*)-5-benzyl-2,2-dimethyl-1,3-dioxolan-4-ol **1**, which in situ generates the corresponding α-hydroxyaldehyde, to achieve the anti-β-allenyl alcohol products (*S*,*R*)-**2**, via the reaction with secondary amines and pinacol allenylboronate. This reaction proceeded under catalyst-free and mild conditions (room temperature), without detectable racemization, showcasing the efficiency of the Petasis MCR (Scheme 2A) [17]. In the quest for the total synthesis of the precursor **4** of legionaminic acid, a bacterial monosaccharide present in the surface of pathogens such as Legionella pneumophila, Seeberger and co-workers also used a chiral α-hydroxyaldehyde **3** as the substrate for a Petasis MCR under mild reaction conditions, which proved to be a key step to attain this natural product (Scheme 2B) [18].

Other α-substituted chiral aldehydes that are suitable to undergo Petasis MCR are the N-protected α-amino aldehydes **5**. This approach was used by Norsikian and co-workers to attain a library of enantioenriched 1,2-trans-diamines (*R*,*S*)-**6** (15 derivatives) in various yields (up to 72%) and enantiomeric excess (the best values were achieved when a nosyl group was used as protecting group (PG), whereas Ac and Boc protected amines lead to poorer enantioselectivities) (Scheme 2C) [19]. A few years later, the same group increased the diversity of the substrates, achieving similar results [20].

Recently, the generation of the enantiomeric pure α-fluoro-aldehydes **7** using (*S*)-α,α-bis[3,5-bis-(trifluoromethyl)phenyl]-2-pyrrolidinemethanol trimethylsilyl ether as an organocatalyst or the (*R*) counterpart and *N*-fluorobenzene-sulfonimide (NFSI) as a fluorinating agent enabled the synthesis of a library of β-substituted-β-fluoroamines **8** (19 examples), in overall moderate to good yields, via the Petasis MCR. The resulting products were suitable to be converted into valuable heterocyclics and, therefore, this approach can depict an interesting approach for synthesizing fluoro-substituted drug candidates, since the inclusion of the fluorine atom often leads to desirable pharmacokinetic profiles (Scheme 2D) [21].

Carbohydrate chemistry is one of the fields where the Petasis MCR has contributed to obtaining high-value molecules from easily accessible building blocks, often prepared using green methodologies from biomass. Pyne and co-workers devoted their attention to the synthesis of alkaloids, such as calystegine B$_4$, DMDP (2*R*,5*R*-di(hydroxymethyl)-3*R*,4*R*-dihydroxypyrrolidine), and DAB (1,4-dideoxy-1,4-imino-D-arabinitol). Similar to many natural products, these alkaloids possess stereocenters and, therefore, their use as building blocks, which induce enantioselectivity, is a straightforward approach. For the synthesis of calystegine B$_4$, (−)-D-lyxose **9** was used as the starting point. This sugar was allowed to react with benzylamine and (*E*)-2-phenylvinylboronic acid to afford aminotetrol **10** in an overall very good yield (82%), while keeping the chiral centers intact. The robustness of this first synthetic pathway step paved the way for the synthesis of this bioactive alkaloid (Scheme 3A) [22]. For the synthesis of DMDP and DAB, L-xylose **11** was the sugar used as the starting material. The designed synthetic route displayed several steps, including the Petasis MCR, which enabled the introduction of the nitrogen atom to the chemical framework, forming the 1,2-anti-amino alcohol intermediate **12**, crucial for the chemical structure of these alkaloids. This synthetic step presented a good yield (76%), without modifying the chirality of the stereocenters (Scheme 3B) [23].

**Scheme 2.** Recent approaches for the Petasis MCR using α-substituted chiral aldehydes as the carbonyl component; (**A**) Synthesis of anti-β-allenyl alcohol products; (**B**) Synthesis of a precursor of legionaminic acid; (**C**) Synthesis of of 1,2-trans-diamines; (**D**) Synthesis of β-substituted-β-fluoroamines.

**Scheme 3.** Petasis MCR as a key-step for the synthesis of natural alkaloids from sugars. (**A**) Synthesis of an aminotetrol, precursor of calystegine $B_4$; (**B**) Synthesis of an 1,2-anti-amino alcohol, a DAB and DMDP precursor. TFE: trifluoroethanol.

Conduramines, structurally aminocyclohexenetriols derived from conduritols, are relevant building blocks for several bioactive and natural products, working as synthetic intermediates of compounds such as alkaloids, aminosugars, and sphingosines [24]. Once again, carbohydrates, namely D-galactose **13**, D-mannose **14,** and D-ribose **15**, proved to be suitable starting materials to achieve the stereoselective synthesis of conduramines. The first two, used by Ghosal and Shaw, enabled the synthesis of (+)-conduramine E and (−)-conduramine E, respectively, in a multistep synthetic approach, where the Petasis MCR played a key role to attain these valuable compounds (Scheme 4A,B) [25]. Norsikian and co-workers reported the use of D-ribose **15** in the synthesis of conduramines **17**, via an unprecedented intramolecular Petasis MCR, as they prepared a dual functionalized intermediate **16**, bearing the α-hydroxyaldehyde and the vinylic boronic acid compounds (Scheme 4C) [26].

The sugar derivative erythorbic acid **18** (also known as D-araboascorbic acid) was employed by Mandai and co-workers and converted into the corresponding chiral lactol derivative **19**, which undergoes Petasis MCR. This strategy enabled the synthesis of a small library of Petasis adducts **20** (8 examples) bearing three adjacent chiral stereocenters in good to very good yields (65–93%) (Scheme 5). These compounds could be easily converted into more functionalized products, showcasing the versatility of the Petasis MCR to achieve structural diversity [27].

The Petasis MCR is also a suitable tool to be employed with other important reactions, increasing and diversifying the molecular structure. The group of Cannillo reported a remarkable example of the application of a synthetic methodology using a domino Petasis/Diels-Alder reaction, enabling the preparation of hexahydroisoindole scaffolds **22**. The enantioselectivity of the synthetic process was triggered by the presence of enantiopure α-hydroxyaldehydes **21**, including different aldoses (Scheme 6). The amine and the boronic acid components required the presence of an alkene and a conjugated diene, respectively, to enable the Diels-Alder step. The resulting library of **22** was obtained with good to excellent yields (54–94%) [28].

(A)

(B)

(C)

**Scheme 4.** Petasis MCR in the synthesis of conduramines, using D-galactose (**A**), D-mannose (**B**), and D-ribose (**C**) as starting materials. TBAF: tetra-n-butylammonium flouride.

**Scheme 5.** D-Erythorbic acid **18** as starting material for the Petasis MCR.

**Scheme 6.** Domino Petasis/Diels-Alder reaction for the synthesis of hexahydroisoindole derivatives. HFIP: hexafluoroisopropanol.

### 2.2. Amine Component

As previously described in this work, the Petasis MCR is a valuable tool to obtain synthetic intermediates of natural products bearing multiple stereocenters. Pyne and co-workers applied the Petasis MCR to obtain a key synthetic intermediate **25** of 9β-hydroxyvertine, a bioactive alkaloid. To achieve this goal, they prepared a chiral primary amine **23** and combined it with a chiral α-hydroxyaldehyde **24** and β-styrenylboronic acid via the Petasis MCR (Scheme 7). This approach led to the formation of a mixture of diastereomers **25** [29]. A similar strategy was used by the same research group to perform a key step in the total synthesis of the natural product hyancinthacine C5, via the Petasis MCR preparation of an anti-1,2-diamino alcohol [30].

**Scheme 7.** Combining a chiral primary amine **23** and a chiral aldehyde **24** to achieve a key synthetic intermediate **25** of the alkaloid 9β-hydroxyvertine.

The easy access to commercially available chiral amines, and in particular chiral 1,2-amino alcohols **26** and their application in the Petasis MCR with glyoxylic acid, enables the synthesis of the valuable heterocycle oxazinone **27**. This strategy has been used by the independent groups of Grajewska and Churches (Scheme 8A) and despite the high yields observed (up to 99%), poor diastereoselectivities in the catalyst-free process were achieved (up to 3:1) [31,32].

The diastereoselectivity was considerably improved when 1,2-amino alcohols **26** were employed in the Petasis MCRs involving other aldehydes other than glyoxylic acid. This strategy has been employed by Huang and co-workers in multiple accounts that used gem-difluoroallylboronates **28** to afford gem-difluotohomoallylamine derivatives (R,S,S)-**29** (12 examples) with great yields (up to 98%) and diastereoselectivities (>99:1) (Scheme 8B) [33]. In more recent accounts, they expanded the scope of boronates used in the Petasis MCR and verified that the reaction could be performed at room temperature with great diastereoselectivity and good yields when using DMSO as a solvent in the presence of methanol (5 equivalents) [34,35].

Another versatile amine component for the development of the asymmetric Petasis MCR is tert-butylsulfinamide **30**. Combining this reagent with several boronic acids and glyoxylic acid enables the generation of enantioenriched β,γ-unsaturated α-amino acids (R,R)-**31** (Scheme 9). While Churches and co-workers reported this strategy for substituted styrenylboronic acids in excellent yields (90–99%) and diastereoselectivity (up to >20:1 dr) [36], shortly after, Li and Xu showed that, when expanding the scope of

the boronic acid to other vinylboronic acids (13 examples), the presence of a Lewis acid, such as InBr$_3$, enabled higher yields [37].

(A)

R$^1$, R$^2$= Bn, Ph, propargyl, Me
R$^3$= C$_2$H$_2$Ph, Ar, Ph

**27**
up to 99% yield
up to 3:1 dr

(B)

R$^1$, R$^2$= H, Ph, Bn
R$^3$= Ph, Ar, cyclohexyl, C$_2$H$_2$Ph, furanyl, thienyl
R$^4$= OTs, Ph, naphthyl

(R,S,S)-**29**
up to 93% yield
up to >99:1 dr

**Scheme 8.** Chiral 1,2-amino alcohols **26** as valuable starting materials for diastereoselective Petasis MCR for the preparation of oxazinones (**A**) and gem-difluotohomoallylamines (**B**). *: stereocenter.

(a) CH$_2$Cl$_2$, rt, 12 h
or
(b) InBr$_3$ (10 mol%)
CH$_2$Cl$_2$, rt, 12 h

R$^1$= C$_2$H$_2$Ar, C$_2$H$_2$R',
benzofuranyl, benzothiophenyl

(R,R)-**31**
(a) up to 99% yield
up to >21:1 dr
(b) up to 78% yield
up to 99% de

**Scheme 9.** *tert*-Butylsulfinamide **30** as the amine component on the Petasis MCR.

The synthesis of α-amino acids using the Petasis MCR is also a relevant strategy, as several post-Petasis reactions unlock access to relevant scaffolds in organic and medicinal chemistry. The group of Bułyszko reported the synthesis of tetrahydroisoquinoline derivatives (*S*)-**33** via the diastereoselective reaction involving a chiral aminoacetaldehyde acetal **32**, a glyoxylic acid, and a boronic acid (Scheme 10). The resulting amino acids were obtained in excellent yields (83%-quantitative) and overall good stereoselectivity (up to 79:21) and could easily undergo hydrogenolysis to afford the N-deprotected amino acid followed by a Pomeranz–Fritsch–Bobbitt cyclization/hydrogenolysis step, affording the final bicyclic N-heterocycle (+)-6,7-dimethoxy-1,2,3,4-tetrahydroisoquinoline-1-carboxylic acid **34** [38].

**Scheme 10.** Synthesis of (+)-6,7-dimethoxy-1,2,3,4-tetrahydroisoquinoline-1-carboxylic acid **34** using the Petasis MCR as a key step.

*2.3. Boronic Acid Component*

The induction of enantioselectivity driven by the boronic acid component is the least explored field in what concerns the asymmetric Petasis MCR. This can be explained due to the mechanistic issues, as the stereocenter is generated at the carbon where the iminium intermediate is formed (see Scheme 1), as well as to the lack of commercially available chiral boronic acids. Over the recent years, only one example can be found in the literature, reported by Crassous and co-workers. They prepared an enantiopure carbo[6]helicenyl boronate (*M*)-**35**, which reacted with morpholine and glyoxylic acid to afford the corresponding Petasis adduct (*M,R*)-**36**, with moderate stereocontrol (7:3 dr) (Scheme 11). In this case, the observed asymmetric induction was controlled by the helicity of the boronic pinacol ester component [39].

**Scheme 11.** Asymmetric Petasis MCR induced by the helical chirality of the boronic acid component (*M*)-**35**.

## 3. Organocatalysts versus Transition-Metal Catalysts

Organocatalysis had demonstrated its competence in the last two decades as a powerful environmentally and economically friendly synthetic tool for reaction processes and valuable reaction products [40,41]. The uses of chiral organocatalysts have had a remarkable track record in inducing enantioselectivity in the reaction product's outcome [42]. Comparatively to the use of the old ("but still gold") chiral transition-metal catalysts, organocatalysts features several advantages from a synthetic and bench handling point of view, such as low toxicity, tolerance to air and moisture, and easy disposal of the chiral building blocks. Although they have been appreciably used as chiral ligands in transition-metal catalysis, several types of organocatalysts have been established over the last few years as efficient organocatalysts and applied successfully in several enantioselective reactions. Chiral diol-, thiourea-, and hybrid thiourea-diol chiral organocatalysts are examples reported in the literature so far and, curiously, applied resourcefully in the asymmetric Petasis MCR, providing great results in terms of enantioselectivity and yields.

As far as we are aware and regarding the literature search, the use of transition-metal catalysts in the asymmetric Petasis MCR is narrowly explored. Very few reports were found considering the use of transition-metal catalysts to promote asymmetry in the reaction between aldehydes (or glyoxylic acid), amines, and boronic derivatives resulting in stereodefined amine products. We highlight the use of palladium catalysts, coordinated or not with chiral ligands, and a curious deviation of the Petasis MCR, where copper catalysts and phosphoramidite-type chiral ligands provided access to chiral *N*-heterobenzyl or benzhydryl amines.

### 3.1. Organocatalysts
#### 3.1.1. Chiral Diol-Based Organocatalysts

Between all the chiral organocatalysts explored in the asymmetric Petasis MCRs, the diol-based catalysts, such as derivatives of BINOL and VANOL are among the most reliable ones for this reaction approach, due to their ability to coordinate with the reaction components, particularly with the iminium intermediate and the boron component (see Scheme 1), facilitating the asymmetric transformation. The group of Schaus reported a pioneering work regarding the use of chiral diol-type organocatalysts to induce asymmetry in the Petasis MCRs [43–47]. Their investigation started by testing several chiral biphenol catalysts in the Petasis MCR using simple alkenyl boronates, secondary amines, and ethyl glyoxylate as components. Several BINOL, VANOL, and VAPOL type catalysts were screened, with the commercially available (*S*)-VAPOL as the best catalyst, giving access to the chiral α-amino ester derivatives (*R*)-**37** in good yields and high enantioselectivities (Scheme 12A) [43]. The reaction showed a good functional group tolerance for amine derivatives (even diamines proved to be good components for this Petasis MCR) and electron-rich and -deficient styrenyl boronates. Remarkably, the catalyst (*S*)-VAPOL can be recovered from the reaction medium and reused without loss of activity and enantioselectivity. In addition, the Petasis adduct **37** can be converted into the related free amine and carboxylic acid in two steps in excellent yields, maintaining the configuration on the chiral center. Earlier mechanistic insights using NMR and ESI-MS analysis highlighted the importance of the glyoxylate ester unit and indicated a single ligand exchange between the catalyst and the boronate. These preliminary studies anticipate the expansion of the Petasis MCR to a diastereoselective variation, successfully reported by the same group a few years after, catalyzed by chiral biphenols using (*S*)-dioxolanol derivatives **1** as the carbonyl component to access enantiopure anti- and syn-β-amino alcohols **38** (Scheme 12B) [44]. The reaction showed high functional group tolerance and isolation of the pure syn-β-amino alcohols (*S*,*R*)-**38** in up to 96% yield. The uncatalyzed reaction using the same reaction components afforded exclusively the anti-diastereomer. L and D-amino acid derivatives, as amine components, were also tested in this reaction approach to verify the outcome of the amino acid configuration on the product diastereoselectivity. In general, the L-amine derivative provides the syn-product (*S*,*S*,*R*)-**39** and the D-amine provides the anti-product (*R*,*S*,*R*)-**39** (Scheme 12C), in moderate to high yields. In addition, a glycolaldehyde dimer was used successfully in this Petasis MCR.

Jiang and Schaus anticipated a methodology, which is highly enantioselective regardless of the imine intermediate, that uses allylboronates as nucleophiles in the Petasis MCR, activating them with chiral diols and biphenols [46]. This one-pot two-step (imine formation followed by allyboration) Petasis allylation reaction is efficient for several aldehydes (aliphatic, heteroaromatic and ethyl glyoxylate) and amines with different electronic and steric properties, giving easy access to chiral homoallylic amine products **40** in excellent yields (up to 99% yield) and enantioselectivities (up to 99:1 er) (Scheme 13). Ideal conditions defined the use of (*R*)-Ph$_2$-BINOL as the best catalyst along with the bench-stable allyldioxaborolane, under microwave heating. Similar reaction conditions were applied to the crotylation reaction of the generated imines, using (*E*)- and (*Z*)-crotyl boronate, proving access to both syn- and anti-diastereomers **40** with two chiral centers (Scheme 13). The reactions were consistent with the Zimmerman–Traxler transition state model, resulting in

high diastereoselectivities. The use of (*E*)-crotyl boronate resulted in the formation of the anti-diastereomer (*R*,*S*)-**40** in high diastereo- and enantioselectivity (>20:1 dr and 98:2 er), whereas the use of (*Z*)-crotyl boronate afforded the syn-diastereomer (*R*,*R*)-**40** in a lower yield and diastereoselectivity for the *para*-methoxy aniline derivative (35 % yield and 9:1 dr) (Scheme 13).

**Scheme 12.** Asymmetric Petasis MCR catalyzed by chiral diol-based organocatalysts, VAPOL (**A**) and BINOL (**B**,**C**) type. TFT: α,α,α-trifluorotoluene.

Thomson and co-workers developed a new one-pot synthesis of allenes based on the Petasis MCR, through the coupling of a hydroxy aldehyde or ketone, sulfonyl hydrazones, and alkynyl trifluoroborate salts [48]. The reaction proceeds with the generation of propargylic hydrazide intermediates by the addition of alkynyl trifluoroborates to sulfonyl-hydrazones. The breakup of the hydrazide intermediates eliminates the spontaneously produced sulfinic acid, producing unstable propargylic diazene intermediates that decompose by a retro-ene reaction to form the corresponding allene products. This reaction approach was pointed out to be a traceless Petasis MCR, affording an allene product instead of the usual amine product. Due to the importance of allenes [49] as versatile building blocks in chemical transformations and their wide distribution in nature, the group of Schaus has looked into the enantioselective version of the traceless Petasis MCR using

chiral diols as catalysts with high levels of asymmetric induction [45]. Two enantioselective procedures were reported to access the chiral allenes **42** and **43** from achiral precursors using the sulfonylhydrazone derivatives **41** via alkynyl boronate addition to glycolaldehyde imine and allylation of alkynyl boronates (Scheme 14). In the first reaction approach, the hydroxy group of the carbonyl component was found to be crucial due to its coordination with the alkynyl boronate components. Glycolaldehyde, $\alpha$-hydroxyacetone, and (*S*)-$\alpha$-hydroxy aldehyde derivatives (for a diastereoselective version) were found to be efficient in this traceless Petasis MCR, using the (*S*)-(CF$_3$)$_4$-BINOL organocatalyst and a mixture of toluene and mesitylene as solvents to improve the reaction selectivity. In the second reaction approach, allyl boronates were added to alkynyl imines to generate the 1,3-alkenyl allenes **43** (Scheme 14). A good reaction scope regarding the use of electron-rich and -deficient aldehyde substrates was accomplished by applying a chiral (*R*)-phenyl-BINOL derivative as the organocatalyst.

**Scheme 13.** Asymmetric Petasis allylation and crotylation reactions catalyzed by (*R*)-Ph$_2$-BINOL organocatalyst.

Studies considering the use of sulfonylhydrazines **41** and allyl boronates in the traceless Petasis MCR were developed by the group of Schaus, extending the scope of the reaction to the use of enals as the aldehyde component, catalyzed by chiral biphenols. The resulting acyclic 1,4-diene products **44**, having either alkyl- or aryl-substituted benzylic stereocenters, were obtained in good to excellent yields and high enantiomeric ratios (Scheme 15) [47]. The reaction occurred via a sigmatropic rearrangement of a transient enantioenriched allylic diazene intermediate and the best reaction conditions were appropriate to a variety of enals, even a silyl methyl derivative. In addition, (*E*)- and (*Z*)-crotylboronate were tested with similar reaction conditions, generating the 1,4-diene products **44** having two methyl-substituted stereocenters in either a syn- or anti-correlation, that highlighted the reaction's diastereoselectivity. Moderate to good yields and high levels of stereocontrol were achieved (Scheme 15).

**Scheme 14.** Enantioselective allene synthesis via a traceless Petasis MCR approach using BINOL-type organocatalysts.

**Scheme 15.** Asymmetric traceless Petasis MCR using enals to access acylic 1,4-diene products **44**.

Yuan and co-workers reported an enantioselective version of the Petasis MCR using salicylaldehydes, amines, and organoboronic acids as reaction components and (*R*)-BINOL as the organocatalyst [50]. Optimization of the reaction conditions disclosed the use of (*R*)-BINOL as the best catalyst and mesitylene as the solvent at 0 °C. The reaction demonstrated very good scope regarding salicylaldehyde derivatives, secondary amines, and vinyl and aromatic organoboronic acids (Scheme 16). A broad range of alkylaminophenol products **45** were obtained in good yields (up to 87%) and moderate to good enantioselectivities (up to 86% ee). A reaction mechanism pathway was proposed by the authors to explain the (*R*)-configuration of the desired product (*R*)-**45**, using (*R*)-BINOL as a catalyst (Scheme 16).

The carbinolamine intermediate was formed by the nucleophilic addition of the secondary amine to the salicylaldehyde derivative. Dehydration of the carbinolamine intermediate affords the key iminium intermediate, which is attacked by the complex formed previously by the organoboronic acid and the BINOL catalyst. This attack is favorable from the *re*-face giving the (*R*)-alkylaminophenol product **45**.

**Scheme 16.** Enantioselective Petasis MCR among salicylaldehydes, secondary amines and organoboronic acids catalyzed by (*R*)-BINOL. A possible reaction pathway.

The research group of Shi reported interesting work considering the binaphthol-catalyzed asymmetric Petasis MCR of salicylaldehydes, secondary amines, and aryl- and vinylboronates [51,52]. In an exhaustive study regarding the reaction parameters, such as boronate scope, solvent, and binaphthol organocatalyst, for the corresponding loading and influence of molecular sieves in the reaction outcome, among others, it was reported that the optimal conditions corresponded to (*S*)-Me-BINOL, toluene as solvent, and room temperature (Scheme 17A) [51]. The presence of 4Å molecular sieves as a water removal agent enhanced the rate of the reaction and also afforded products (*R*)-**45** with up to 99% ee. Comparable effects were observed with 3Å molecular sieves and $MgCl_2$ (as $Na_2SO_4$ demonstrated no effect on yield and enantioselectivity). Cyclic and acyclic secondary amines provided the corresponding products in good yields and enantioselectivities, except for $HNCy_2$. Relative to the uncatalyzed reaction, the authors conclude that the binaphthol-catalyzed pathway was generally 500 times faster for generating products with 99% ee. NMR [51] and DFT calculations [52] afforded insights into the origin of the accelerated rate of the reaction and the mechanism outcome. (*R*)-BINOL and vinyl boronates were used to explain the reaction mechanism (Scheme 17B). It was determined that BINOL accelerates the rate-determining step by forming an energetically favorable cyclic hemiaminal complex with the hemiaminal intermediate, which undergoes the formation of a cyclic iminium complex by losing a water molecule. A hydrogen bond between the iminium cation and the oxygen from the BINOL articulates the stereochemistry of the vinyl group's migration, directing it to the *re*-face of the iminium species. Due to ring strain, BINOL is rapidly released via ligand exchange to regenerate the catalyst and the chiral amine product (*R*)-**45** is formed (Scheme 17B). This mechanism is extended for the use of vinyl and aryl

boronates in the Petasis MCR, since the (R)-BINOL provides selectivity for the *re*-face addition products for both components. Benzaldehyde is unreactive under the same reaction conditions, demonstrating the importance of the *ortho*-hydroxy group in this reaction pathway.

(A)

R$^1$= H, 4-Br, 5-NO$_2$, 4-OMe
R$^2$, R$^3$= aliphatic cyclic and acyclic, morpholine
R$^4$= C$_2$H$_2$R'

(B)

**Scheme 17.** BINOL-catalyzed Petasis MCR of vinyl and aryl boronates (**A**) and the reaction mechanism (**B**).

Using a similar (R)-BINOL derived catalysts, Mao and co-workers reported an enantioselective version of the Petasis MCR using aliphatic and aromatic amines, ethyl glyoxylate, and alkenyl and heteroaryl trifluoroborate salts (long-term stability). The corresponding heterocyclic-derived α-amino esters **46** were obtained in moderate to good yields (up to 82% yield) and enantioselectivities (up to 82% ee), in mild reaction conditions (Scheme 18) [53]. By thin-layer chromatography reaction monitoring, traces of the imine intermediate were found, indicating that it was not completely consumed during the reaction, even for long reaction times. Several additives were tested to overcome this issue and activate the trifluoroboronate salts. Together with molecular sieves, LiBr proved to be the best choice to achieve good yields and promising enantioselectivity. Electron-rich trifluoroborate salts (such as thiophene, pyrrole and indole) afforded the best yields and enantioselectivities. When N-boc substituted indoles and pyrrole were used as the boron component, the corresponding products were accomplished with low enantioselectivities. Aromatic and aliphatic amines were also tested successfully in this reaction approach, although the reaction only works using ethyl glyoxylate as the carbonyl component.

**Scheme 18.** BINOL-type-catalysed enantioselective synthesis of heterocyclic-derived α-amino esters **46** through Petasis MCR with trifluoroborate salts.

In the last years, our group has been active in the synthesis of privileged heterocyclic scaffolds with druglike properties for applications in medicinal chemistry [54–58]. Based on the creation of new sustainable processes with high atom- and step-economy, we establish the efficacy of the BINOL-catalyzed Petasis MCR in the design of key highly substituted-oxindole [59] **49** and tryptanthrin [60] **50** derivatives in high yields and enantioselectivities (Scheme 19). Using salicylaldehyde derivatives, secondary amines, and arylboronate substituted key scaffolds (oxindole **47** or tryptanthrin **48**), it was possible to access great structural diversity in good yields, enantioselectivities, and diastereoselectivities. The (*R*)-BINOL-catalyzed Petasis enantioselective reaction works well in a gram-scale, giving access to 5-α-substituted-oxindole benzylamine derivatives (*R*)-**49** up to 99% yield and up to 99% ee, which were easily converted into the resultant isatin-type scaffold with biological interest [59] We postulate, based on the literature precedents, that a nucleophilic attack on the *re*-face by the iminium intermediate is likely to occur, affording the desired products with an (*R*)-configuration in case of the 5-α-(3,3-disubstituted oxindole)-benzylamine derivatives [59] (*R*)-**49** and (*S*)-configuration in case of the tryptanthrin derivatives [61] (*S*)-**50**. Moderate fungicidal bioactivity was revealed by the library of tryptanthrin derivatives (*S*)-**50**.

Yan and co-workers reported very recently a synthetic methodology to access axially chiral asymmetric biaryltriols with broad functional group tolerance under mild reaction conditions and several useful applications [62]. Beyond proving their value as selective fluorescent sensors toward the Ru$^{3+}$ ion, they were also used successfully as chiral ligands for the asymmetric preparation of chiral sec-alcohols and as organocatalysts in the Petasis MCR. Salicylaldehyde derivatives, secondary amines, and dibutyl vinylboronates were used in the Petasis MCR approach, catalyzed by an unsymmetric biaryltriol organocatalyst in mild reaction conditions (Scheme 20). Excellent yields (92–95%) and high enantioselectivities (up to 83% ee) were achieved for the Petasis adducts (*S*)-**45**.

**Scheme 19.** (*R*)-BINOL-catalysed asymmetric Petasis MCR in accessing valuable scaffolds with promising biological applications.

**Scheme 20.** Enantioselective Petasis MCR catalysed by an unsymmetric biaryltriol organocatalyst.

### 3.1.2. Chiral Thiourea-Based Organocatalysts

Chiral urea/thiourea have been recognized as efficient organocatalysts in a plethora of asymmetric transformations, particularly in asymmetric cost-effective and environmentally benign MCRs [63]. Their unique double hydrogen bonding capacity and resulting ability for the dual activation of both the electrophile and nucleophile simultaneously in the chemical reaction, triggering remarkable enantioselectivity, has made these bifunctional organocatalysts very popular in organic synthesis in the last years [64–66]. Ground-breaking work reflecting the application of these organocatalysts in the asymmetric Petasis MCR was developed by the group of Takemoto [67–70]. They describe the first catalytic version of the Petasis MCR using quinolines as the amine component and a newly designed thiourea organocatalyst ([ThUr Cat], Scheme 21) [67]. The reaction exhibits a good scope regarding quinoline and boronic acid derivatives, being electron-rich boronic acids more reactive. Together with the thiourea-catalyst, a combination of water and NaHCO$_3$ was used as an additive to obtain higher stereoselectivity. The regeneration of the catalyst was assumed to be promoted by a proton source and the removal of the resulting boronic acid by the base [66]. A 1,2-amino alcohol functionality on the thiourea organocatalyst was crucial

for the outcomes of this Petasis MCR. It was proposed by the authors that the 1,2-amino alcohol functionality activates the boronic acid and the thiourea unit providing sufficient activation of the *N*-phenoxycarbonyl quinolinium salt by double H-bonding (see Scheme 21). Excellent enantioselectivities (up to 97% ee) for the 1,2-Petasis adducts (*R*)-**51** were accomplished with a thiourea-amine organocatalyst having a chelating functionality. It should be highlighted that only 10 mol% of the organocatalyst was used for loading.

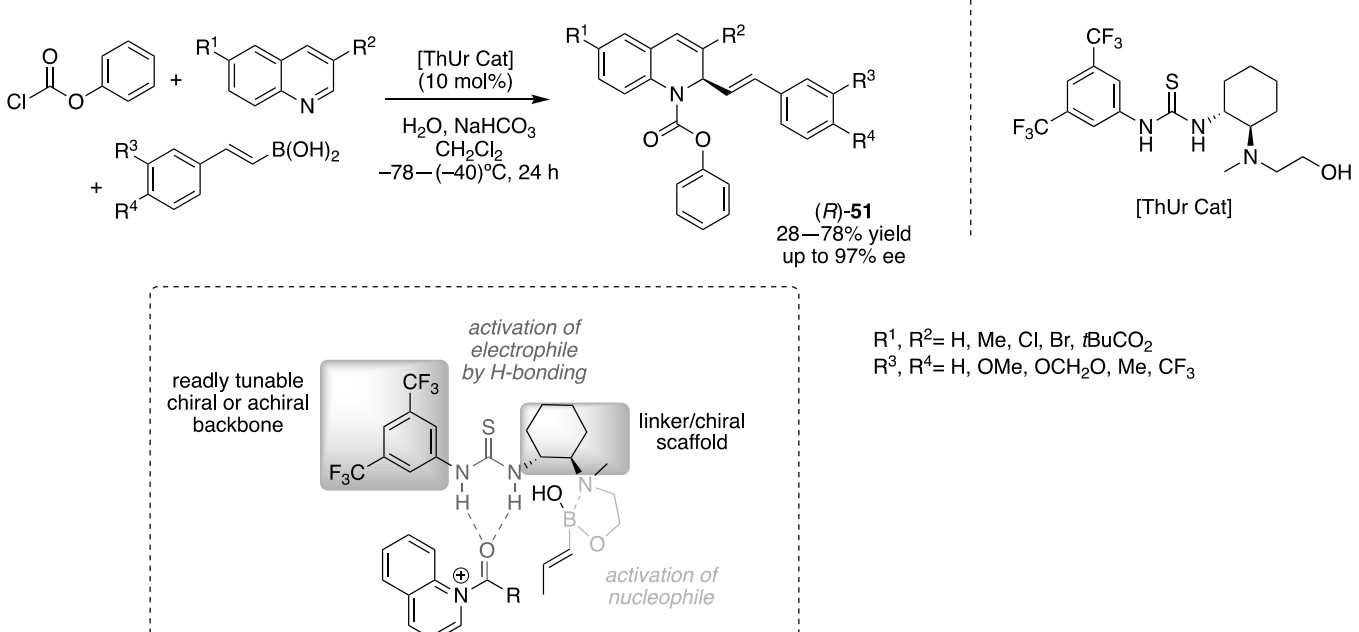

**Scheme 21.** Thiourea-catalysed enantioselective Petasis MCR of quinolines and the proposed activation mode.

The generation of these chiral thiourea-type organocatalysts motivated the authors to explore further their use in the Petasis MCR. The synthesis of optically active *N*-aryl amino acid derivatives (*S*)-**53** was reported using a bifunctional hydroxy thiourea organocatalyst [ThUrO Cat] in the two-step Petasis MCR between *N*-aryl-α-imino acids **52** and vinyl boronates (Scheme 22) [68]. The introduction of another Lewis basic site into the carbon linker of the organocatalyst facilitates the effective formation of a complex between the organoboronic acid and the catalyst. The ether moiety in the thiourea catalyst proved to be more efficient in this reaction outcome. The reaction runs in two steps, the first being the formation of the not isolated imine derivative **52** (Scheme 22). The reaction presents a good scope regarding boron components and aniline derivatives. Electron-rich vinyl boronates afforded good results with excellent enantioselectivities. Aniline derivatives containing hydroxymethyl or amino groups were also suitable. This two-step Petasis MCR can be applied also to the synthesis of peptide oligomers and the adducts that were interconverted into heterocyclic structures of pharmaceutical interest.

**Scheme 22.** Thiourea-organocatalyzed enantioselective synthesis of *N*-aryl amino acid derivatives (*S*)-**53** through the two-step Petasis MCR.

### 3.1.3. Miscellaneous

Due to the tremendous success achieved by diol- and thiourea-type organocatalysts in the Petasis MCR, the group of Yuan decided to design a new hybrid thiourea-BINOL organocatalyst [ThUr-BINOL] and test its efficiency in the Petasis MCR among salicylaldehydes, amines, and organoboronic acids [71]. The reaction protocol can be extended to a wide variety of substrate components, generating a library of alkylaminophenols (*R*)-**45** in moderate to high yields (48–92%) and excellent enantioselectivities (up to 95% ee) (Scheme 23). Electron-donating and withdrawing substituents in the aldehyde components showed good activities and enantioselectivities in this reaction outcome. In addition, a sterically constrained aldehyde was used, but with low yield and enantioselectivity for the reaction adduct. Cyclic amines were the most efficient components for this reaction protocol, and a variety of aryl- and vinylboronic acids were used successfully. A gram-scale version of the reaction was effective in demonstrating the potential of this method. A possible activation mode was reported by the authors, in which the thiourea binds with the phenol anion of the iminium intermediate. Simultaneously, a cyclic BINOL-derived boronate ester fragment is formed by the exchange of the diol group with the hydroxy groups of the boronic acid. The substituent ($R^4$, Scheme 23) of the boronic ester favorably attacks the iminium ion from the *re*-face, affording the desired adduct **45** with the (*R*)-configuration (Scheme 23). The reaction fails for benzaldehyde derivatives (without an *ortho*-hydroxy group), primary amines, acyclic secondary amines, and alkyl boronic acids.

**Scheme 23.** Enantioselective Petasis MCR catalyzed by a thiourea-BINOL organocatalyst. MTBE: methyl *tert*-butyl ether.

### 3.2. Transition-Metal Catalysts

In 2007 and to the best of our knowledge, the group of Szabó was the first to report interesting studies regarding the use of pincer-complex-palladium catalysts in the Petasis MCR [72]. Essentially, they explored the application of highly functionalized allyl boronates in several carbon–carbon bond formation reactions addressing the typical issues usually distinguished in the use of this type of reagents, such as the formation of isomeric mixtures, poor group tolerance, stability of the reagents, etc. The Petasis MCR, traditionally a three-component approach, was converted by this group into a four-component derivative, involving the *in situ* generation of the allyl boronate component from freshly prepared pincer-complex-palladium catalysts (Scheme 24). Using commercially available diboronic acids as the boronate source, the first step of this transformation comprises an efficient borylation procedure, where the allyl boronate species was formed in smooth reaction conditions. The amine derivatives and glyoxylic acid were added after 4 to 16 h, depending on the allyl alcohol substrate used. The $\alpha$-amino acid derivatives (*R*)-**54** (11 examples) were formed as single regio- and stereo-isomers from commercially and economically favored allyl alcohol substrates. Mechanistic aspects demonstrated by the authors suggested that the four-component Petasis MCR takes place by at least three different processes: the borylation of the allyl alcohol, the imine formation between the amine derivative and glyoxylic acid, and allylation between the imine intermediate and the allyl boronate species. The amine component (and the glyoxylic acid) cannot be added at the beginning of the Petasis MCR due to possible coordination of the nitrogen atom to the palladium catalyst leading to its deactivation and, thus, preventing the formation of the allyl boronate. The $\alpha$-amino acid derivatives (*R*)-**54** are useful drug intermediates (homologues of natural amino acids) and precious building blocks in API synthesis.

**Scheme 24.** Stereo and regio-defined $\alpha$-amino acid (R)-**54** synthesis via one-pot four-component Petasis MCR from functionalized allyl boronates generated by pincer-palladium complex catalysts.

Almost ten years after, Manolikakes and co-workers reported interesting work regarding the palladium-catalyzed enantioselective Petasis MCR [73,74]. They found earlier that a dual catalyst system comprising a Lewis acid (Yb(OTf)$_3$) and a transition-metal catalyst (Pd(TFA)$_2$ and 2,2'-bipyridine) was a promising choice to obtain access to $\alpha$-substituted amides from readily available amides, aryl aldehydes, and aryl boronic acids (Petasis 3-component approach) in great scope and moderate to good yields [75]. Years after, they successfully developed the enantioselective version of this reaction, due to the importance of chiral $\alpha$-arylamines in biologically active natural products and APIs. After extensive screening, a highly enantioselective Petasis MCR was developed using Pd(TFA)$_2$ as a catalyst, in combination with an easily accessible chiral bis(oxazoline)-ligand, from sulfonamides, aldehydes, and arylboronic acids (Scheme 25A) [73]. The incredible scope was achieved with this catalytic system using several sulfonamides, aldehydes (aryl and alkyl substituted), and arylboronic acids. Despite generally excellent enantioselectivities (up to 99:1 er) for almost all the tested components and moderate to excellent yields, the reaction time is very high (64 h) at 40 °C. Remarkably, this method displays high tolerance toward air and moisture, contrary to the conventional use of transition-metal catalysts. The asymmetric Petasis MCR was performed in screw-type vials under an air atmosphere without prior purification of reaction components or solvent. The authors also found out that deprotection of tosyl group from the final $\alpha$-substituted amides (R)-**55** was easily achieved with Na/naphthalene, in moderate yield, affording the free amine product with complete retention of configuration.

Similar to chiral amines, $\alpha$-arylglycines are key units also found in relevant natural products and drugs and an important class of unnatural or nonproteinogenic $\alpha$-amino acids. Manolikakes and co-workers efficiently extended their protocol to access chiral $\alpha$-substituted amides (R)-**55** (Scheme 25A) to achieve $\alpha$-arylglycines (R)-**56** with an excellent level of enantioselectivity (Scheme 25B). Using the same reaction conditions, the difference relies on using glyoxylic acid derivatives (instead of aldehydes) as a component [74]. Despite a good scope regarding sulfonamides and boronic acid derivatives, only three glyoxylic acids were tested with good yield and excellent enantioselectivity regarding the corresponding $\alpha$-arylglycines (R)-**56**. Despite long reaction times, the method has once more proved its effectiveness and robustness for the general derivatives tested (up to 86% yield and up to >99:1 er). Removing common sulfonyl groups (such as tosyl or nosyl) from the N-sulfonyl-protected $\alpha$-arylglycines (R)-**56** can be quite challenging since when using basic conditions or reduction methodologies the $\alpha$-stereocenter could racemize [76]. The authors pointed out the successful use of the 2,2,4,6,7-pentamethyl-2,3-dihydrobenzofuran-5 sulfonyl (Pbf) group in the sulfonamide component, to enable consequent cleavage with an acid (TFA for instance) affording the free amine with complete retention of configuration.

**Scheme 25.** Palladium-catalyzed enantioselective Petasis 3-component synthesis of α-substituted amides (*R*)-**55** (**A**) and α-arylglycines (*R*)-**56** (**B**).

Recently, the same group reported an interesting variation of the palladium-catalyzed Petasis MCR regarding the synthesis of α-arylglycines where the boronic acid component was replaced by benzoic acid derivatives [77]. Despite being a more sustainable version of the classical Petasis MCR, no reports were found in the literature regarding an enantioselective version, so the work is beyond the scope of this review and will not be discussed here.

Zhou and co-workers recently reported an interesting approach to the synthesis of chiral benzylic and benzhydryl amines (*S*)-**58** using the copper complexes of monodentate phosphoramidite-type ligands (PPM) in the asymmetric arylation of *N*-azaarylaldimines **57** with worktop stable arylboroxines [78]. Despite not being considered a Petasis MCR approach in its classical form, we decided to include it in this review since the established method can be applied to the use of aldehyde derivatives, 3-picolyl-2-amine derivatives, and boroxine derivatives in a two-step synthetic approach, without isolation of the corresponding *N*-heteroaryl aldimines **57** (Scheme 26). Using only 1 mol% of the cheapest copper catalyst, together with the spiro-1,1′-diindanyl phosphoramidite ligand (PPM ligand, Scheme 26), it was possible to obtain a library of benzyl and benzhydryl amines (*S*)-**58** in excellent enantioselectivities (up to 97% ee) and very good yields (up to 92% yield). Heteroaryl amines derived from pyrazine, pyrimidine, quinoline, pyrazole, 2-indazole, and 2-benzoisoxazole can be used successfully in this reaction approach showing great reaction scope. Despite high temperatures and long reaction times, the use of copper as a transition-metal catalyst and stable boroxine derivatives as the boron component are the main advantages of this chemical transformation.

**Scheme 26.** An interesting two-step copper-catalyzed asymmetric Petasis MCR approach reaction, in the synthesis of chiral amines (*S*)-**58**.

## 4. Conclusions and Future Perspectives

Three decades have passed since the discovery of the Petasis MCR by Nicos A. Petasis and I. Akritopoulou. Although widely explored in its classical and non-classical versions, this three-component reaction involving organic amines, aldehydes and organoboronic reagents still displays serious limitations, such as specific substrate components, among others. The use of aldehydes containing boron-directed groups, electron-rich alkenyl boron or aryl boronic acids, and specific nucleophilic amines showcased some examples. Despite considerable efforts and great scope regarding the Petasis MCR, in the asymmetric version, these problems still endure, and we found that the literature is still focused on the use of a few classes of catalysts and chiral components. Most of the successful examples reported so far have been Petasis MCRs involving chiral carbonyl substrates and chiral amines. Only two general organocatalyst classes were explored with success in this asymmetric transformation, and the use of transition-metal catalysts still needs to be explored in the future.

The importance of the enantioselective synthesis of chiral amines makes the asymmetric Petasis MCR a direct and effective method for drug discovery and development and other applications of interest in medicinal chemistry, organic chemistry, and chemical biology. Post-Petasis reaction approaches and subsequential functional group modifications are powerful tools nowadays and a great deal of development in these fields is expected in the years to come.

The stereocontrol of the Petasis MCR is still very challenging, and the full potential of this reaction remains unfulfilled. As far as we know, no reports regarding the use of heterogeneous catalysis were found in the literature. This might constitute the next great challenge in asymmetric Petasis MCRs, as it could enhance this reaction's applicability and eco-friendliness.

**Author Contributions:** Conceptualization, writing, review, and editing by C.M. and P.B. All authors have read and agreed to the published version of the manuscript.

**Funding:** C.M. thanks the financial support from PT national funds from Fundação para a Ciência e Tecnologia/Ministério da Ciência, Tecnologia e Ensino Superior (FCT/MCTES) reference UIDB/50006/2020 | UIDP/50006/2020. P.B. acknowledges the financial support by Fundação para a Ciência e a Tecnologia (FCT), Portugal in the scope of the projects UIDB/04565/2020 and UIDP/04565/2020 of the Research Unit Institute for Bioengineering and Biosciences—iBB, and LA/P/0140/2020 of the Associate Laboratory Institute for Health and Bioeconomy—i4HB, and UIDB/04585/2020 of the Research Unit CiiEM.

**Data Availability Statement:** Not applicable.

**Conflicts of Interest:** The authors declare no conflict of interest.

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
