# Peer review of "The Asymmetric Petasis Borono-Mannich Reaction: Insights on the Last 15 Years"

_catalysts, doi:10.3390/catal13061022_

Round 1

Reviewer 1 Report

Dear Authors

I have found your review article very interesting with high potential for publishing, but I want you to do some edition please:

1. In case of fingolimod and zanamivir in fig 1, please bring the synthetic pathway and indicate the step using petasis reaction.

2. In all schemes and figures, please write the reaction condition and reagents above the arrows.

3. For all figures and schemes, please use the same font size.

Author Response

We appreciate the feedback and we totally agree with the comments made by reviewer #1, as these changes will improve the overall quality of this manuscript. Therefore, we:

1. added the synthetic pathway of the mentioned APIs

2. added the reaction conditions and reagents where these were missing and;

3. revisited the font size in all figures and schemes.

Reviewer 2 Report

This review work has the right to be. The language of presentation is clear and topic is interest to a broad audience, primarily, organic and medicinal chemists. However, presented reactions occur without any catalysts, so manuscript is outside scope of Catalysts Journal.

I recommend transfer in other journal (Molecules or IJMS).

Some additional recommendations:

1. The citation style should be unified

2. In some schemes (for example, Scheme 4, 8, 9, 17B) product yields or reaction conditions are missed.

The language of presentation is clear, but there are sevral typos 

Author Response

We appreciate the feedback given by reviewer #2. We understand the point of view given by reviewer 2, concerning the reactions without catalysts. However, in order to increase the visibility of this review, and provide an overall perspective over asymmetric synthesis of Petasis adducts, we believe it is of the interest of the Journal, as well as from the scientific audience, to have this section of the manuscript for comparative purposes.

  1. We revisited the citation style and made it uniform throughout the manuscript.
  2. We revisited the original papers and added the information which was missing.

Reviewer 3 Report

This this article by C. S. Marques and P. Brandão, is a review paper dealing with the advances on the asymmetric version of the so-called Petasis Borono-Mannich over the last 15 years. The work has been divided into 2 main Chapters based on the asymmetric strategy used, such as the exploitation of the chiral pool (sugars) instead of metal- vs organocatalytic systems.

The Petasis Borono-Mannich reaction is an interesting and useful transformation, scarcely addressed by previous review papers: for this reason, this reviewer sustains that this paper will be of interest for the organic chemistry community and, if properly improved, will merit publication in a journal of value such as Catalyst.

Despite the contents of this work are very interesting, the main flaws reside in the graphics. I strongly believe that all Schemes and Figures in a review paper must be self-consistent and easily interpreted. The reader must understand the described reactions and their course or mechanism without having to flip from one page to another to find the yields and the stereoselectivity of the processes; or, even worst, feel obliged to download the original document to understand something about it.

On these bases, I strongly suggest that the authors improve their work with the following adjustments:

a. A proper numbering should be use in the text and graphics to address each molecular entity represented in the work: it will be much easier for the reader to match a reaction with the proper comment in the text.

b. Every reaction in the schemes should report yields, dr’s and ee’s matching those described in the text.

c. Every reaction should report a proper caption in which the nature of the substituents is described. As an example, R1 = Alk, Ar; R2 = Bn, Me, …;

d. If the reaction is diastereoselective, proper descriptor should be used: 3:1 dr (syn/anti)

e. If a reagent or a product is chiral and enantiopure, a proper stereochemical descriptor describing its absolute configuration should be used: (R)-1; (M)-2; (±)-3 …..and so on…

Moderate editing of English language required.

Author Response

We appreciate the feedback given by Reviewer #3 and we agree that the changes requested will improve the overall manuscript quality and readability. With that in mind,

  1. a) we added numbers to the graphics and text to address the products of the reactions. This approach was made in line with the recent reviews published in this journal.
  2. b) We added the information requested where it was missing, after revisiting the original papers.
  3. c) We added the information requested.
  4. d) We added the information requested.
  5. e) We added the information requested.

Round 2

Reviewer 2 Report

The authors agree that the described reactions do not proceed under catalytic conditions. So I still recommend transfer in other journal (Molecules or IJMS), because the manuscript is outside scope of Catalysts.

Author Response

We appreciate the feedback given by reviewer #2. However, what we affirmed after the 1st round of revisions was that indeed we present asymmetric synthesis based on chiral pool reagents, but that concerns a small section of the manuscript. The focus is mostly given to catalytic asymmetric transformations, which falls within the scope of this journal. Furthermore, as stated before, we believe it is of the interest of the Journal, as well as from the scientific audience, to have this section of the manuscript for comparative purposes. As it was not an issue raised by the remaining reviewers or the editor, we keep our submission in the journal Catalysts.

Reviewer 3 Report

Dear sirs,

I’ve read with interest the revised version of this review article by C. S. Marques and P. Brandão, dealing about the advances on the asymmetric version of the Petasis Borono-Mannich reaction over the last 15 years.

The work has been improved and it’s now almost in shape for publication in Catalyst.

Please revise carefully all the text in which persist many typos errors: all Schemes and Figures in brackets have a double bracket at the end such as (Scheme 1)). Numbers should in brackets only when preceded by the corresponding IUPAC- or common names, otherwise they should be indicated without brackets. Also in the graphics, numbers should be without brackets.

Minor editing of English language required

Author Response

We appreciate the feedback given by Reviewer #3 and we agree that the changes requested will improve the overall manuscript quality and readability. We removed the double brackets and the brackets from the numbers corresponding to the structures throughout the text and figures/schemes. We also performed another thorough typo verification and corrected all the mistakes we could find.